# **Evaluating Long-Term Effectiveness of Managed Aquifer Recharge** for Groundwater Recovery and Nitrate Mitigation in an Overexploited Aquifer System

Yuguang Zhu<sup>1,2</sup>, Zhilin Guo<sup>1,2,\*</sup>, Sichen Wan<sup>1,2</sup>, Kewei Chen<sup>2</sup>, Yushan Wang<sup>3</sup>, Zhenzhong Zeng<sup>1,2</sup>, Huizhong Shen<sup>1,2</sup>, Jianhuai Ye<sup>1,2</sup>, Chunmiao Zheng<sup>4</sup>

- <sup>1</sup>State Key Laboratory of Soil Pollution Control and Safety, School of Environmental Science and Engineering, Southern University of Science and Technology, Shenzhen, China
- <sup>2</sup>MEE Key Laboratory of Integrated Surface Water Groundwater Pollution Control, School of Environmental Science and Engineering, Southern University of Science and Technology, Shenzhen, China
- 3Center for Hydrogeology and Environmental Geology Survey, China Geological Survey, Baoding, China 4School of the Environment and Sustainable Engineering, Eastern Institute of Technology, Ningbo, China

Correspondence to: Zhilin Guo (guozl@sustech.edu.cn)

Abstract. Managed aquifer recharge (MAR) has been widely recognized as an effective strategy for groundwater restoration and has been implemented globally. In the North China Plain, over-extraction of groundwater has led to a continuous decline in water levels, forming one of the world's most significant groundwater depressions. Recent riverine MAR operations have shown significant local groundwater recovery, yet the regional-scale hydrological and geochemical impacts of sustained MAR remain insufficiently understood. Most existing studies rely on short-term field monitoring and emphasize localized responses. This study, focusing on Xiong'an depression area, develops a coupled flow and multi-component reactive transport model to evaluate the long-term impacts of MAR on groundwater recovery and the spatiotemporal evolution of water quality. The results indicate that MAR substantially accelerates groundwater recovery and mitigates the regional depression, though the central funnel exhibits a delayed response due to its distance from recharge sources. Nitrate reduction is dominated by dilution effects from recharge water rather than denitrification, with heterogeneity exerting strong control on the spatial pattern but limited influence on overall concentration levels. These findings highlight the dual hydrological and geochemical benefits of sustained MAR and provide quantitative insights for optimizing large-scale recharge strategies in overexploited aquifer systems.

#### 25 1 Introduction

Groundwater is a vital global freshwater resource, playing a crucial role in drinking water supply, agricultural irrigation, and industrial development, particularly in arid and semi-arid regions (Kuang, 2024; Ma, 2024). However, rising demands from population growth, agricultural intensification, and urban expansion have led to unsustainable extraction, with agricultural irrigation alone consuming over 40% of global groundwater withdraws (Guo et al., 2015; Zhang et al., 2016; Gong et al., 2018; Mukherjee et al., 2021). The long-term over-extraction has caused widespread aquifer depletion and significant declines in groundwater levels, leading to severe ecological and environmental consequences, such as seawater intrusion, river desiccation,

https://doi.org/10.5194/egusphere-2025-5534 Preprint. Discussion started: 19 November 2025

© Author(s) 2025. CC BY 4.0 License.

and land subsidence (Peters et al., 2022; Levy et al., 2021; Escriva-Bou et al., 2020). Amid these challenges, groundwater quality degradation, specially salinization and nitrate pollution, has become as a pressing concern in many arid regions (Pauloo et al., 2021; Zhan et al., 2025). These issues are particularly acute in agricultural areas such as the North China Plain (NCP), where the largest groundwater depression in the world has developed since the 1970s (Zheng and Guo, 2022; Wu et al., 2024), resulting in severe ecological deficits and a negative groundwater balance (Zhao et al., 2019; Su et al., 2021).

Managed aquifer recharge (MAR) has been widely adopted as a strategy to mitigate groundwater depletion. MAR methods, including enhanced infiltration, bank filtration, and well injection, primarily use surface water sources to replenish aquifers (Bagheri-Gavkosh et al., 2021; Zhan et al., 2024), with infiltration-based approaches being most common (Sprenger et al., 2017). Global analyses suggest that MAR has contributed to measurable groundwater restoration in about 16% of stressed aquifer regions (Jasechko et al., 2024). However, alongside hydraulic benefits, MAR can trigger geochemical changes that influence groundwater quality (Guo et al., 2023b; Guo et al., 2023a). For example, mineral dissolution during recharge has been shown to increase fluoride and phosphate (Schafer et al., 2018), while oxidized recharge water has induced pyrite oxidation, mobilizing metals (Vergara-Saez et al., 2024; Chen et al., 2023b). Such outcomes underscore that MAR-driven water quality changes are strongly dependent on aquifer mineralogy, recharge chemistry, and biogeochemical conditions.

A critical but less understood dimension is the role of aquifer heterogeneity and biochemical reactions in controlling water quality during MAR. Preferential flow paths in heterogeneous media govern solute transport and reaction zones (Zheng and Gorelick, 2003; Chen et al., 2024), yet regional-scale assessments explicitly integrating these effects remain scarce. While soil-scale and laboratory experiments have quantified nitrogen adsorption, nitrification, and denitrification (Mekala and Nambi, 2016; Zhan et al., 2024), they often omit microbial degradation of organic matter and carbon–nitrogen coupling. Similarly, one-dimensional or small-scale numerical models (Liang et al., 2024) capture aspects of nitrogen transformation under MAR but cannot represent complex groundwater-surface water interactions or long-term cumulative pollution risks. Evidence indicates that dilution effects dominate MAR-induced nitrate decreases (Guo et al., 2023a), but the broader role of heterogeneity and biogeochemical feedback in governing nitrate persistence or removal at regional scales remains unresolved.

Therefore, this study aims to address these gaps by developing a numerical modeling framework to evaluate the long-term impacts of sustained MAR implementation on groundwater recovery and nitrate dynamics in the groundwater depression area. Specifically, we would answer two main questions: 1) How does MAR affect groundwater recovery in a severely depleted aquifer system. 2) How do heterogeneity and biogeochemical reactions interact with MAR to control the spatiotemporal evolution of nitrate concentrations in groundwater.

https://doi.org/10.5194/egusphere-2025-5534 Preprint. Discussion started: 19 November 2025

© Author(s) 2025. CC BY 4.0 License.

# 2 Methodology

### 2.1 Study site

80

- The study area is located in the central Xiong'an New Area, Hebei Province, covering ~536 km². It is bounded by the Nanjuma River (northeast), Baigouyin River Diversion Canal (east), Baiyangdian Lake (south), Dingxing County (northwest), and a shallow groundwater depression cone (west) (Fig. 1). The region lies at the center of the NCP, characterized by flat alluvial plains (average elevation 10–15 m) and Quaternary deposits ranging from 10 to 500 m in thickness. Groundwater depth ranges from 5 to 30 m. Groundwater in the Xiong'an New Area occurs in Quaternary alluvial—pluvial unconsolidated deposits forming four aquifer groups (I–IV). We focus on the shallow system (Groups I–II) and the unsaturated zone. Land use is dominated by arable land, with construction land as a secondary use, and some grassland in certain regions (Fig. 1). The study area has a warm temperate humid monsoon climate, with mean annual precipitation of 400~501 mm, more than 75% of which occurs between June and September.
- Groundwater recharge sources mainly include infiltration from atmospheric precipitation, lateral recharge, river infiltration and irrigation return, while pumping is the dominant discharge. Long-term over-extraction has formed a distinct groundwater depression funnel, typical of the NCP. Since 2018, artificial recharge has been implemented through diversion of South-to-North Water Transfer Project inflows into local rivers, aiming to restore groundwater levels and secure long-term water supply for Xiong'an.

Groundwater and river water samples were previously collected to characterize water quality and chemistry and measure chemical parameters (Chen, 2022). Ca<sup>2+</sup> in groundwater is between 9.075×10<sup>-4</sup> to 2.7×10<sup>-3</sup> mol L<sup>-1</sup>, primarily from the dissolution of calcite and gypsum. HCO<sub>3</sub><sup>-</sup> concentration of groundwater was from 3.1×10<sup>-3</sup> to 8.92×10<sup>-3</sup> mol L<sup>-1</sup> and SO<sub>4</sub><sup>2-</sup> concentration of groundwater was from 3.13×10<sup>-4</sup> to 2.41×10<sup>-3</sup> mol L<sup>-1</sup>. At some sampling points, nitrate concentrations reached 1.94×10<sup>-3</sup> mol L<sup>-1</sup>, exceeding the U.S. Environmental Protection Agency (EPA) drinking water NO<sub>3</sub><sup>-</sup> standard of 8.87×10<sup>-4</sup> mol L<sup>-1</sup>.

Figure 1. Study Area Location and Land Use Distribution. (a) Geographic location of the study site in China, showing the elevation data (DEM) with high and low points. (b) Land-use map of Rongcheng, illustrating the distribution of arable land, forest land, grassland, residential land, water areas, and observation wells.

## 2.2. Numerical model

# 2.2.1. Governing equation

95 Groundwater flow and reactive transport under MAR were simulated using PFLOTRAN (Hammond et al., 2014), a massively parallel 3D reactive transport model. Variably saturated flow is described by the Richards equation:

$$\frac{\partial}{\partial t}(\varphi s \eta) + \nabla \cdot (\eta \mathbf{q}) = Q_w , \qquad (1)$$

$$\mathbf{q} = -\frac{k\rho_w g}{\mu_w} \nabla (P - \rho_w g z) , \qquad (2)$$

120

where  $\varphi$  is porosity; s is saturation;  $\eta$  is the mole water density [kmol m<sup>-3</sup>]; q is Darcy's flow velocity (positive: upwelling) [m s<sup>-1</sup>];  $Q_w$  is the source/sink term for mass transport [kmol m<sup>-3</sup> s<sup>-1</sup>)]; k is intrinsic permeability [m<sup>2</sup>];  $\rho_w$  is the water density [kg m<sup>-3</sup>];  $\mu_w$  is the water viscosity [kg m<sup>-1</sup> s<sup>-1</sup>)]; k is the gravitational constant [m s<sup>-2</sup>]; k is pressure [Pa]; k is the vertical component of position vector (positive: upwelling) [m].

The governing equation for multi-component reactive transport in single-phase flow is,

$$105 \quad \frac{\partial}{\partial t} (\varphi s \Psi_i) + \nabla \cdot \mathbf{\Omega}_i = Q_i - \sum_m v_{jm} I_m - \frac{\partial s_i}{\partial t}, \tag{3}$$

$$\mathbf{\Omega}_{i} = (q - \varphi s D \cdot \nabla) \Psi_{i} , \qquad (4)$$

where *i* represents the *i*th primary species hereinafter,  $\Psi_i$  is the total concentration [mol L<sup>-1</sup>],  $\Omega_i$  is the total flux [mol L<sup>-1</sup> s<sup>-1</sup>].  $Q_i$  is the source/sink term [mol L<sup>-1</sup> s<sup>-1</sup>],  $v_{jm}$  is the stoichiometry coefficient of *i*th primary species in mineral m,  $I_m$  is the mineral precipitation/dissolution rate based on transition state rate [mol L<sup>-1</sup> s<sup>-1</sup>], and  $S_i$  is the sorbed concentration [mol L<sup>-1</sup>].

Denitrification was modeled as a two-step reduction ( $NO_3^- \to NO_2^- \to N_2$ ) (Table 1), with acetate ( $CH_3COO^-$ ) as the electron donor, following Michaelis–Menten kinetics (Chen et al., 2023a; Chen et al., 2025):

$$R = \mu_{max} \frac{c_{ED}}{c_{ED} + K_{ED}} \frac{c_{TEA}}{c_{TEA} + K_{TEA}} \frac{K_1}{K_1 + C_1},\tag{5}$$

where  $\mu_{max}$  is the rate constant [mole L<sup>-1</sup> s<sup>-1</sup>];  $C_{ED}$ ,  $C_{TEA}$  and  $C_I$  represent the concentrations of the electron donor (ED), terminal electron acceptor (TEA), and inhibitor [I) [mol L<sup>-1</sup>];  $K_{ED}$ ,  $K_{TEA}$  and  $K_I$  are the half-saturation constants for the electron donor, acceptor, and inhibitor [mol L<sup>-1</sup>]. It should be noted that multiple inhibitors may simultaneously affect specific redox reactions. For example, according to the redox gradient theory, sulfate reduction may be inhibited by oxygen, nitrate, and trivalent iron dissolved in groundwater. These effects are modeled using an inhibition term in equation (5). The parameters for reactive transport are provided in Table S1.

Table 1 Two-step denitrification

| Reaction                                                                           | Rate                                                                                                                                      |  |  |
|------------------------------------------------------------------------------------|-------------------------------------------------------------------------------------------------------------------------------------------|--|--|
| $CH_3COO^- + 4NO_3^-$<br>→ $2HCO_3^- + 4NO_2^- + H^+$                              | $R_{NO_3^-} = \mu_{max} \frac{[NO_3^-]}{[NO_3^-] + K_{TEA}} \frac{[CH_3COO^-]}{[CH_3COO^-] + K_{ED}} \frac{K_{I,O_2}}{K_{I,O_2} + [O_2]}$ |  |  |
| $CH_3COO^- + 2.667NO_2^- +$ $1.667H^+ \rightarrow 2HCO_3^- +$ $1.33N_2 + 1.33H_2O$ | $R_{NO_3^-} = \mu_{max} \frac{[NO_2^-]}{[NO_2^-] + K_{TEA}} \frac{[CH_3COO^-]}{[CH_3COO^-] + K_{ED}} \frac{K_{I,O_2}}{K_{I,O_2} + [O_2]}$ |  |  |

#### 2.2.2 Model setup

The model domain (33 km × 26.9 km × 135 m) was discretized into 100 m horizontal grids and three vertical layers: 15 m, 40 m, and 80 m, representing the vadose zone, the phreatic aquifer, and the underlying unit, respectively. The permeability coefficients of the aquifer were determined through pumping tests. Using hydrological and geological data from Hebei Province, lithofacies paleogeographic features, and previous parameter zoning maps, a permeability coefficient zoning map (Fig. 2a) was generated, providing the baseline scenario (homogeneous fields) for the flow model (Table S2). Additionally, a fully heterogeneous scenario was simulated to evaluate impact of heterogeneity, especially preferential flow path on the performance of MAR and its effects on groundwater quality. Four lithofacies: clay, silt, fine sand, and medium-to-coarse sand were identified by 14 borehole lithologies (Fig. S1) and used to conditionally generate heterogeneous field using Transition Probability Geostatistical Software (*T-PROGS*), which is based on transition probability—Markov chain model to generate 3D realizations of random fields (Carle and Fogg, 1997). To represent structural uncertainty, we generated 20 TPROGS-conditioned heterogeneous realizations and simulated each realization independently; results were summarized by the ensemble mean across realizations. One representative heterogeneous realization is shown (Fig. 2b); the full parameter list is provided in the Appendix (Table S3). The unsaturated zone was modeled using the van Genuchten–Mualem model to describe the relationship between soil water content and soil water potential, calculating both saturation and relative permeability.

Figure 2. Model parameterization and Initial condition. (a) Permeability zoning for the baseline (homogeneous) field. (b) Lithofacies-conditioned heterogeneous permeability field generated with TPROGS (one realization shown). (c) Spatial Distribution of Initial Head (2017).

The flow boundary conditions are assigned by incorporating the physical conditions on each side. The upper boundary is set as a flow boundary driven by daily precipitation. Additionally, since the area is agricultural, there are high concentrations of

145 nitrates in the surface layer from irrigation water. The bottom bounded by bedrock is set as a no flow boundary. The western boundary coincides with the groundwater-depression margin and is imposed as a specified-head (Dirichlet) boundary using the initial potentiometric surface, The southern boundary, corresponding to Baiyangdian Lake, is assigned a constant head of 6.5 m. The northern and eastern boundaries are designated as recharge flow boundaries, with fluxes determined from monitoring reports (Jin et al., 2024). The measured groundwater flow field in 2017 was adopted as the initial condition for the 150 model (Fig. 2c). Initial and boundary solute conditions for reactive transport were derived from measured chemical conditions of recharge water and groundwater (Table 2).

The artificial recharge scheme is designed based on the field measurements of flow data from the Xingaifang station during 2018-2019. The recharge period is defined as August to December each year. The simulation assumes that the recharge duration, location, and infiltration efficiency remain constant over the prediction period. and the evolution of groundwater levels and solute concentrations is projected until 2035.

Table 2 Compositions of MAR water and groundwater

| Species                          | MAR water (mol L-1)   | Groundwater (mol L-1) |
|----------------------------------|-----------------------|-----------------------|
| NO <sub>3</sub>                  | 7.86×10 <sup>-5</sup> | 4.28×10 <sup>-4</sup> |
| CH <sub>3</sub> COO <sup>-</sup> | 7.96×10 <sup>-5</sup> | 5.84×10 <sup>-5</sup> |
| $O_2(aq)$                        | 1.48×10 <sup>-4</sup> | 5.2×10 <sup>-5</sup>  |
| $NO_2^-$                         | 1×10 <sup>-8</sup>    | 1×10 <sup>-8</sup>    |
| $N_2$                            | 1×10 <sup>-8</sup>    | 1×10 <sup>-8</sup>    |
| рН                               | 7.65                  | 7.43                  |

### 2.3. Simulation cases

160 Eight simulation scenarios (Table 3) were designed to evaluate the impact of geological structural conditions, artificial recharge measures, and denitrification processes on nitrate concentration dynamics in groundwater. In the groundwater flow simulation, a homogeneous case was established by dividing the model domain into four hydrogeologic zones based on local geological records (Li et al., 2023), with the hydraulic conductivity of each zone determined from pumping test results (Xu, 2022) to represent the regional hydraulic properties of the study area. To investigate the influence of stratigraphic heterogeneity on solute migration, reactive transport simulations were performed for 20 heterogeneous realizations and the ensemble-averaged 165 results were compared with those from the homogeneous case.

Scenario combinations are constructed by considering the presence or absence of artificial recharge, denitrification, and geological heterogeneity, thereby enabling a comprehensive assessment of individual and interactive effects. To assess the

impact of artificial recharge on regional groundwater level recovery, Case 1 and Case 2 are compared, while the role of artificial recharge in nitrate removal is analyzed by comparing Case 1 with Case 2 and Case 5 with Case 6 under different geological structures. The denitrification contribution is analyzed by comparing Case 1 with Case 3 and Case 5 with Case 7 to quantify the role of denitrification in nitrate reduction. The synergistic effect is discussed by comparing Case 1 with Case 4 and Case 5 with Case 8 to assess the combined impact of artificial recharge and denitrification reactions and their additive or antagonistic effects.

To reduce bias caused by concentration magnitude difference among scenarios and to highlight relative changes, the concentration differences ( $\Delta C$ ) for each scenario are logarithmically transformed. This transformation helps visualize the distribution patterns of high and low variation zones in the spatial distribution analysis.

Table 3 Simulation cases

| Case | Geological Conditions | MAR | Denitrification |
|------|-----------------------|-----|-----------------|
| 1    | Homogeneous           | No  | No              |
| 2    | Homogeneous           | Yes | No              |
| 3    | Homogeneous           | No  | Yes             |
| 4    | Homogeneous           | Yes | Yes             |
| 5    | Heterogeneous         | No  | No              |
| 6    | Heterogeneous         | Yes | No              |
| 7    | Heterogeneous         | No  | Yes             |
| 8    | Heterogeneous         | Yes | Yes             |

## 3 Result

185

190

180

# 3.1 Model calibration

To verify the model's reliability, simulated groundwater levels were compared against observations from four observation wells (Point 1–Point 4) over 2017–2019 (Fig. 3a–d). The results show that the simulated water level trends at the four observation points are generally consistent with the measured values (R<sup>2</sup> = 0.95) and accurately reflect the dynamic changes in water levels during both the recharge and non-recharge periods. In particular, during the first recharge phase from August to December 2018, rise in water levels was observed at Point 1 and Point 2, which are located near the recharge area, and the model was able to accurately capture this sudden increase. In contrast, the more muted responses at distal wells (Points 3 and 4) were also reproduced. This indicates that the constructed groundwater flow model reliably represents the spatiotemporal

variations in groundwater levels under recharge and non-recharge conditions, providing a robust hydrodynamic basis for solute transport analysis (Fig. 3e).

Figure 3. Model calibration results: (a)—(d) Simulated and observed water level time series at observation points. (e) Correlation coefficient between simulated and observed values.

# 3.2 Impact of MAR on groundwater levels

We assess basin-scale groundwater level responses to MAR during 2023–2035 by comparing simulations with and without recharge (Fig. 4). Without MAR, the cone of depression persists with limited natural recovery (Fig. 4a–d). MAR systematically elevates water levels, weakens the depression, and propagates higher heads inward from the river margins (Fig. 4e–h). The corresponding difference field (Δh, defined as head MAR – head<sub>no MAR</sub>), shows a persistent high-Δh corridor along the northeastern river margin in all periods (Fig. 4i–l), with maximum rises up to 7.5 m and a mean gain of 1.11 m. Influence extends ~5 km from the recharge reach, decaying toward the depression center where recovery remains limited. From January 2023 to January 2031, the high-Δh belt expands slightly inland and the gradient relaxes (Fig. 4i–k); by January 2035 (Fig. 4l), the pattern stabilizes with only modest additional propagation. Taken together, MAR elevates heads most strongly near the river and transmits effects inward with distance- and connectivity-controlled attenuation, consistent with the area's permeability structure (Fig. 4).

Figure 4. Spatial distribution of groundwater levels and changes with and without MAR (a)–(d) Spatial distribution of groundwater levels without MAR. (e)–(h) Spatial distribution of groundwater levels with MAR. (i)–(l) Spatial distribution of groundwater head differences.

Time-series head from five monitoring points (Fig. S2) further illustrate distance- and permeability-dependent responses Points 1 and 2 are located near the recharge area, with Point 1 in a high-permeability zone and Point 2 in a low-permeability zone. Points 3 and 4 are progressively farther from the recharge location, with the distance between Point 4 and the recharge location being approximately twice that of Point 3. Point 5 lies within the cone of depression. Near the river, Point 1 (high-permeability) and Point 2 (low-permeability) exhibit rapid onsets during recharge windows and persistent head gains thereafter. Although the two sites respond nearly synchronously, their amplitudes diverge with permeability: Point1 rises by ~4–7 m, while Point2 increases by ~3–5 m across the projection horizon (Fig. 5a–b). Superimposed on the step-like elevation is a distinct seasonal cycle—peaks that coincide with recharge pulses and troughs that reflect background abstraction—indicating that MAR augments, rather than overrides, the natural intra-annual variability.

Figure 5. Groundwater level changes and differences at observation points; (a)–(e) Time series of water level changes at observation points (Point 1 to Point 5) with and without MAR; (f) Differences in water levels (MAR vs Without MAR) for all observation points

With increasing distance from the river, the MAR signal becomes weaker and arrives later. Inland points (Points 3 and 4) exhibit delayed and dampened responses ( $\sim 2-4$  m and  $\sim 1-2$  m, respectively), while the depression-center well (Point 5) shows

only minor gains. Seasonal cycles remain evident at all sites, with recharge pulses superimposed on background abstraction. The amplitude hierarchy (Point 1 > 2 > 3 > 4 > 5) is consistent across years, demonstrating that permeability controls recovery magnitude near the river, while distance and connectivity govern inland attenuation.

### 235 3.3. Individual effect of MAR and Denitrification on nitrate distribution

Under homogeneous geology, MAR induces a smooth and continuous nitrate reduction pattern, strongest along the river corridor and extending inland over time, (Fig. 6a–d). From 2023 to 2035, the high-reduction belt expands landward while preserving strong spatial continuity and year-to-year persistence. By 2035, cells exhibiting a discernible reduction cover 67.85% of terrestrial model cells, indicating that in homogeneous media the recharge signal propagates rapidly and relatively uniformly and sustains a basin-scale decline in nitrate.

Figure 6. Nitrate reduction due to MAR (without reaction). (a–d) Homogeneous geology, 2023/2027/2031/2035; (e–h) Heterogeneous geology, same periods.

In contrast, under heterogeneous condition, the same reduction appears, but it is strongly patchy and anisotropic (Fig. 6e–h). High-permeability corridors form reduction hotspots, while low-permeability and poorly connected zones show weaker or negligible changes. Inland expansion is faster and more fragmented compared to the homogeneous case with 73.43% of cells affected by 2035, 5.58% higher than under homogeneous conditions, and pronounced reductions remain concentrated within ~15 km of the river. This contrast reflects permeability-controlled flow partitioning and residence times that high-permeability pathways accelerate dilution and flushing, whereas low-permeability zones restrict response.

The spatial responses can be summarized by area-contribution analysis of signed log-difference classes relative to the baseline (Fig. 7a-c). Negative values indicate decreases in nitrate concentration; within this negative range, values closer to zero represent larger absolute decreases. The response classes correspond to orders of magnitude: -8 to -7 (very small decrease), -7 to -6 (small), -6 to -5 (moderate to large), and > -5 (the strongest decreases, at least on the order of  $10^{-5}$ ). Cells labeled NaN are lie below the magnitude threshold. Both homogeneous and heterogeneous settings show a growing footprint of meaningfully responding cells (non-NaN) and a shift toward stronger response classes, but heterogeneity case achieves greater magnitude. In the homogeneous field (Fig. 7a), the non-responding cells (NaN) declines from 39.0% (2023) to 32.2% (2035), the moderate-large class (-6 to -5) expands from 5.4% to 19.6%, and the strongest class (> -5) from 1.28% to 3.28%, while the small-reduction class (-7 to -6) remains modal at roughly 31-33%. In the heterogeneous field (Fig. 7b), the same pattern holds, but the magnitude is enhanced: NaN decreases from 35.2% to 26.57%, and the expansions into -6 to -5 and > -5 reach 22.43% and 3.79%, respectively, bigger gains than in the homogeneous case (Fig. 7c). Thus, MAR consistently expands the footprint of nitrate reductions, while heterogeneity enhances the intensity of reductions but weakens their spatial uniformity. To isolate the role of denitrification, recharge was disabled and only the reaction process was simulated. Under homogeneous geology, denitrification yields a spatially continuous nitrate decrease in nitrate across the basin (Fig. 8a-d), which intensifies over time but retains a uniform pattern, indicating reaction control. Under heterogeneous geology, the reduction remains domain-wide but slightly non-uniform, with near-river high-permeability zones showing weaker declines due to shorter residence times that limit reaction (Fig. 8e-h).

Figure 7. Area-contribution analysis of concentration log-difference response classes under MAR. (a) Homogeneous; (b) Heterogeneous; (c) Relative difference = (homogeneous – heterogeneous) / heterogeneous. Bars show area fractions at four snapshots (2023, 2027, 2031, 2035).

Figure 8. Nitrate reduction due to reaction (without MAR). (a-d) Homogeneous; (e-h) Heterogeneous.

Area-contribution analysis confirms this systematic trend (Fig. 9a–c). Both homogeneous (Fig. 9a) and heterogeneous (Fig. 9b) fields show a coherent order-of-magnitude shift and remain closely aligned in magnitude. The domain evolves from being entirely 

Figure 9. Area-contribution analysis of concentration log-difference response classes under denitrification. (a) Homogeneous; (b) Heterogeneous; (c) Relative difference.

# 3.4. Combined effect of MAR and Denitrification with Spatial Mean Analysis

When MAR and denitrification act together, nitrate declines basin-wide relative to the no-recharge, no-reaction baseline (Fig. 10). Under homogeneous geology (Fig. 10a–d), the combined effect produces a smooth near-river belt that expands inland from 2023 to 2035, yielding a broad footprint with 95.58% of terrestrial cells affected. The spatial pattern indicates recharge-driven dilution dominates within ~5 km of river, where concentrations drop rapidly, while denitrification governs the slower decay in the 5–15 km inland zone. Under heterogeneous geology (Fig. 10e–h), the combined effect also lowers nitrate but forms a patchy, anisotropic mosaic due to preferentially flow along high-permeability corridors. The overall affected fraction is 94.38%, and spatial heterogeneity is more evident than in the homogeneous case, with stronger contrasts between well-connected and poorly connected regions.

Area-class analysis (Fig. 11a–c) shows that the combined forcing yields the largest and most persistent expansion of favorable classes in both settings, with the heterogeneity case advancing further. Under the homogeneous field (Fig. 11a), the small-reduction band (-7 to -6) increases from 49.8% in 2023 to 60.3% in 2035, the moderate–large band (-6 to -5) from 6.0% to 27.7%, and the strongest band (>-5) from 1.28% to 3.31%. In the heterogeneous field (Fig. 11b), the small-reduction band is nearly unchanged, while the moderate–large (2.9% to 31.0%) and strongest (1.35% to 3.83%) bands grow more than in the homogeneous case. Unlike the MAR-only runs, the heterogeneous setting also retains a larger share of non-responding cells (NaN increasing from 2.96% to 5.61%, compared with 1.67% to 4.42% in the homogeneous case). Overall, dilution coupled with reaction maximizes nitrate decrease; the homogeneous configuration secures greater gains in the moderate–large class, whereas heterogeneity acts as a brake on the strongest tail and accentuates spatial unevenness.

Figure 10. Nitrate reductions with recharge and denitrification (a-d) Homogeneous; (e-h) Heterogeneous.

Figure 11. Area-contribution analysis of concentration log-difference response classes under combined MAR and Denitrification. (a) Homogeneous; (b) Heterogeneous; (c) Relative difference.

Along the transect at y = 13km, four monitoring points (x = 28, 25, 22 and 19 km, listed from nearest to farthest from the recharge reach) were used to evaluate the MAR effect as the concentration difference between the MAR and no-MAR scenarios, where positive values indicate that MAR lowers nitrate and larger values denote stronger reductions. In the homogeneous field (Fig. 12a), all sites exhibit step-like year-to-year increases that align with the August–December recharge period, evidencing a clear multi-year cumulative effect; spatially, the response decays monotonically with distance from the recharge reach (28 km > 25 km > 22 km > 19 km), indicating distance control under near-uniform media. In the heterogeneous field (Fig. 12b),

330

335

The near-river site (28 km) shows the largest amplitudes, with higher post-season steps and faster cumulative growth than under homogeneity. the two mid-distance sites (x = 25 and 22 km) have nearly identical onset times and amplitudes, implying that preferential high-permeability pathways transmit the recharge signal and offset simple distance control. By contrast, the distal site (x = 19 km) has a weak mean response but an uncertainty band second only to the near-river site (28 km), indicating large variability in far-field connectivity across realizations. Overall, heterogeneity amplifies the near-field response while flattening differences between the two mid-distance sites, and increases spatiotemporal variance-patterns consistent with the area-class analysis.

Figure 12. Time series of MAR and no-MAR concentration difference ( $\Delta C$ ) at monitoring points along y = 13km: (a) homogeneous permeability; (b) heterogeneous permeability.

Average concentration analysis indicates that by 2035, nitrate reductions from recharge alone, denitrification alone, and their combined effect are  $2.55 \times 10^{-6}$ ,  $2.62 \times 10^{-7}$  and  $2.81 \times 10^{-6}$  mol L<sup>-1</sup>, respectively, under homogeneous conditions, and  $2.75 \times 10^{-6}$ ,  $2.61 \times 10^{-7}$  and  $3.01 \times 10^{-6}$  mol L<sup>-1</sup> under heterogeneous conditions (Fig. 13a). These correspond to nitrate reductions of

30.72%, 3.16%, and 33.8% in homogeneous, and 31.73%, 3.01%, and 34.75% in heterogeneous settings (Fig. 13b). These findings suggest that: (i) recharge plays a dominant role in nitrate removal, accounting for about 91% of the total reduction, significantly more than denitrification (about 9%); (ii) geological heterogeneity slightly enhances the recharge effect (1.01%), but has little to no impact on denitrification (p > 0.05); (iii) the combined effect is nearly equal to the sum of the individual effects (difference 

Figure 13. Domain-mean nitrate response over time. (a) Absolute difference in concentration relative to the baseline scenario (mol  $L^{-1}$ ). (b) Relative change (%).

#### 4 Discussion

This study quantitatively evaluates the impact of MAR on groundwater level recovery and water quality evolution in the groundwater depression area of Xiong'an New Area by using a three-dimensional groundwater flow and solute transport model. The study found that MAR significantly alleviated the groundwater depletion issue, particularly near rivers, where water levels rose rapidly; while the improvements in the central depression area are more modest due to its distance from the recharge area. The reduction in nitrate concentration is mainly attributed to the dilution effect of recharge water, while denitrification had a minimal contribution. Geological heterogeneity significantly influenced the spatial distribution of nitrate distribution, with high-permeability pathways accelerating dilution but exerting little influence on denitrification.

These findings align with previous studies (Li et al., 2019; Zhang et al., 2020), which emphasize the importance of artificial recharge in groundwater recovery. Our study advances this understanding by integrating nitrate biogeochemical reactions with regional-scale flow dynamics, quantitatively distinguishing the contributions of dilution and denitrification that often simplified in earlier modeling efforts (Guo et al., 2023a). Moreover, long-term simulations highlight both the potential and the limits of MAR for sustainable groundwater management, showing that while MAR is effective for alleviating depletion and reducing nitrate concentrations, its benefits diminish with distance and are constrained by persistent agricultural inputs. he pronounced rise in water levels near rivers, contrasted with limited recovery in cone centers, highlights the critical role of hydraulic connectivity and suggests that strategically distributed recharge interventions may be more effective. However, the limited availability of organic electron donors and the generally oxidizing shallow aquifers, which constrain microbial nitrate reduction, resulted in minor contributions from denitrification. Although heterogeneity enhances recharge effectiveness by promoting preferential flow, it does not necessarily favor redox reactions, since rapid flow reduces residence times and limits opportunities for denitrification.

Despite these insights, certain limitations remain. Biogeochemical reactions were simplified to two-step denitrification, omitting other potentially important processes such as assimilation into microbial biomass, transformation into ammonia, which are strongly controlled by oxygen availability, organic carbon, and microbial communities (Mosley et al., 2022; Wang and Li, 2024). The recharge scheme assumed constant conditions, whereas in practice recharge efficiency and timing are likely to vary under different management or climatic scenario (Kourakos et al., 2019; Liu et al., 2024). Furthermore, the complex vertical interactions between groundwater and recharge water were also simplified, especially the dynamic changes of organic matter and nitrogen in deep groundwater, which could also impact nitrate spatial distributions (Levintal et al., 2023; Seibert et al., 2016).

Future research should strengthen long-term field monitoring of both groundwater levels and solute concentrations to refine model calibration and validation. Expanding the representation of nitrogen transformation pathways and organic matter cycling

https://doi.org/10.5194/egusphere-2025-5534 Preprint. Discussion started: 19 November 2025

© Author(s) 2025. CC BY 4.0 License.

EGUsphere Preprint repository

will enable more realistic predictions of groundwater quality. Additionally, MAR optimization strategies should be evaluated under different hydrological and geological conditions to enhance the effectiveness of water quality and water level recovery. Furthermore, integrating socio-economic factors into MAR will be essential for developing more actionable MAR implementation plans, ensuring their systemability and wide applicability.

implementation plans, ensuring their sustainability and wide applicability.

**5 Conclusions** 

405

A coupled groundwater flow and solute transport model was used to assess MAR effects in the Xiong'an depression cone, focusing on water-level recovery and nitrate dynamics. Simulations show that MAR markedly alleviates groundwater depletion near rivers, with diminishing effects toward the depression center. Nitrate reductions are primarily driven by dilution from recharge water, while denitrification provides a secondary contribution. Geological heterogeneity shapes the spatial variability of nitrate decreases by channeling flow through preferential pathways, which enhance dilution but do not substantially alter reaction processes.

Overall, MAR is shown to be an effective management tool for stabilizing water levels and mitigating nitrate pollution in heavily exploited aquifers, though its performance is spatially variable and limited in distal depression zones. The findings highlight both the potential and the constraints of MAR as a long-term strategy, offering valuable guidance for designing regionally adapted recharge operations.

Data availability

Data used in this article are available from the authors upon reasonable request.

400 CRediT authorship Contribution Statement

Yuguang Zhu: Conceptualization, Investigation, Methodology, Software, Visualization, Writing – original draft. Zhilin Guo: Conceptualization, Formal analysis, Methodology, Supervision, Funding acquisition, Resources, Writing – review & editing. Sichen Wan: Formal analysis, Writing – review & editing. Kewei Chen: Formal analysis, Methodology, Writing – review & editing. Yushan Wang: Formal analysis, Writing – review & editing. Zhenzhong Zeng: Supervision, Resources, Writing – review & editing. Huizhong Shen: Formal analysis, Writing – review & editing. Jianhuai Ye: Formal analysis, Writing – review & editing. Chunmiao Zheng: Writing – review & editing.

## **Declaration of Competing Interest**

The authors declare no conflict of interest.

# Acknowledgments

This study was supported by the National Natural Science Foundation of China (42377045), the Guangdong Provincial Basic and Applied Basic Research Fund (2024B1515020038), High Level University Special Funds (G03050K001), and the Center for Computational Science and Engineering of Southern University of Science and Technology.

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
