# Peer review of "Evaluating Long-Term Effectiveness of Managed Aquifer Recharge for Groundwater Recovery and Nitrate Mitigation in an Overexploited Aquifer System"

_EGUsphere, 2025_

## Community Comment (CC2)

**#CC1**

The manuscript presents a 3D variably saturated flow and multi-component reactive transport model (PFLOTRAN) to evaluate the long-term impacts of managed aquifer recharge (MAR) on groundwater recovery and nitrate mitigation in the Xiong'an depression, North China Plain. You explicitly separate the contributions of dilution and denitrification and explore the role of geological heterogeneity using T-PROGS-based realizations.

The topic is timely and important, the study area is of high practical relevance, and using a regional 3D reactive transport model for long-term MAR evaluation is scientifically interesting. The overall narrative is clear and the paper is generally well organized. I recommend publication after considering these comments:

**Response:**

Thank you for your encouraging feedback on the value of this work. We greatly appreciate your insights and recommendations, which have significantly improved the clarity and depth of our research. In response to your comments, we have made the necessary revisions. Below, we provide a point-by-point response to each comment in the revised manuscript.

1. The TPROGS-based heterogeneous fields are central to the conclusions, but the conditioning data (only 14 boreholes), transition probabilities, variograms, and convergence of 20 realizations are not sufficiently documented. Please add a dedicated subsection describing the limitations of representing heterogeneity with such sparse conditioning.

   **Response:**

   We thank the commenter for bringing this vital point to our attention. We agree that the limited conditioning data constrain the TPROGS-based heterogeneous fields. In the revised manuscript, we explicitly acknowledge that the TPROGS realizations should be regarded as plausible but non-unique representations of heterogeneity, used primarily to explore the sensitivity of regional MAR performance to facies connectivity rather than to provide deterministic cell-scale predictions. We will highlight this limitation when interpreting heterogeneity-related results.

   Revisions have been made in line 380:

   "The geological heterogeneity was characterized based on a sparse borehole dataset, inevitably introducing structural uncertainty in the delineation of localized contaminant migration, although its impact is partially mitigated through stochastic ensemble simulations."

2. The description of pumping, irrigation returns, and lateral boundary conditions is quite brief relative to their importance, and there is no explicit groundwater or nitrate mass balance. Please provide a brief summary in the paper.

   **Response:**

   We thank the commenter for this suggestion. These boundary conditions were already described in lines 146–155 of the original manuscript, and we elaborate on them here for clarity.

Groundwater pumping rates were primarily compiled from public datasets and were further calibrated against observed water-level variations in regional monitoring wells to better represent the actual extraction stress in the Xiong'an New Area. Because the study area is predominantly agricultural, irrigation return flows were estimated based on published regional studies. For lateral boundaries, the western boundary was prescribed as a specified-head (Dirichlet) boundary using the initial potentiometric surface, while the southern boundary adjacent to Baiyangdian Lake was treated as a constant-head boundary. The eastern boundary was prescribed as a recharge (flux) boundary, with fluxes constrained using observed discharge data from nearby surface-water gauging stations. A groundwater water-balance summary is provided in the Supplementary Information (Table S4).

Revisions have been made in line 190:

"A domain-wide groundwater budget was computed to support the interpretation of simulated flow dynamics, and the major inflow and outflow components are summarized in the Supplementary Information (Table S4)."

Table S4. Groundwater water-budget

| mass (kg) | Storage change | Pre | Irrigation | western | southern | eastern | Total pumping |
|---|---|---|---|---|---|---|---|
| $T_{end} - T_{initial}$ | $1.69 \times 10^{11}$ | $5.22 \times 10^{11}$ | $1.43 \times 10^{11}$ | $5.04 \times 10^{11}$ | $5.02 \times 10^{11}$ | $3.98 \times 10^{11}$ | $-1.9 \times 10^{12}$ |

3. The introduction outlines gaps but does not clearly state what this work adds beyond existing MAR–nitrate modeling in the NCP and globally. Please sharpen the problem statement, explicitly contrast your framework and findings with key prior regional-scale MAR studies, and clearly articulate the unique methodological and management contributions in the last paragraph of the Introduction. I strongly recommend to consider "Assimilation of sentinel-based leaf area index for modeling surface-ground water interactions in irrigation districts"

**Response:**

We thank the commenter for this insightful suggestion. We agree that, in its current form, the Introduction does not yet clearly convey what our study adds beyond existing MAR–nitrate modeling work in the North China Plain and globally. In the revised manuscript, we will sharpen the problem statement and more explicitly contrast our framework with key regional-scale MAR studies, emphasizing that our work (i) develops a fully 3D variably saturated flow and multi-component reactive transport model for a large groundwater depression cone, (ii) explicitly quantifies the relative roles of dilution and denitrification using domain-integrated nitrate mole balances, and (iii) evaluates the impact of geological heterogeneity through ensembles of T-PROGS realizations. We will also consider and cite the recommended study on assimilation of Sentinel-based leaf area index for modeling surface–groundwater interactions in irrigation districts, and position our work as complementary to that line of research.

Revisions have been made in line 62:

"This paper is organized as follows. First, we describe the hydrogeological setting, water budget, and

groundwater quality of the study area. We then present and justify the modeling framework (calibrated flow, reactive transport for nitrate, and T-PROGS–based heterogeneity), followed by the scenario results. Finally, we discuss management implications for MAR, key limitations, and the transferability of the approach to other overexploited aquifer systems."

4. The conclusion that ~91% of nitrate reduction is due to dilution is based primarily on domain-average concentration differences between scenarios. Please support this attribution with explicit nitrate mass-balance terms (advective–dispersive fluxes vs. reaction sinks).

**Response:**

We thank the commenter for this insightful suggestion. To address this concern, we conducted a full-domain nitrate mass balance analysis between the initial and final simulation states and explicitly quantified the contributions of advective–dispersive transport (dilution) and biogeochemical reactions.

Table R2 summarizes the nitrate mass balance over the simulation period. The total change in nitrate mass within the domain is $-8.723 \times 10^7$ mol. This net reduction can be decomposed into three components: (i) nitrate inflow associated with recharge and boundary fluxes ($+1.254 \times 10^9$ mol), (ii) nitrate outflow through groundwater discharge and pumping ($1.333 \times 10^9$ mol), and (iii) nitrate removal via denitrification reactions ($-8.25 \times 10^6$ mol).

Based on this explicit mass balance, advective–dispersive transport processes account for approximately 91% of the total nitrate reduction, while denitrification contributes only about 9%. This quantitative result is fully consistent with the concentration-based analysis presented in the main text, but provides a more rigorous and physically grounded attribution of nitrate attenuation mechanisms.

Table R1. Nitrate mass balance between initial and final simulation states

| $\Delta NO_3^-$ mass (mol) | Global | Inflow | Outflow | Reaction |
|---|---|---|---|---|
| $T_{end} - T_{initial}$ | $-8.723 \times 10^7$ | $1.254 \times 10^9$ | $-1.333 \times 10^9$ | $-8.25 \times 10^6$ |

Revisions have been made in line 339:

"To rigorously verify the mechanisms driving these reductions, a domain-wide mass balance was computed, revealing a net nitrate storage change of $-8.723 \times 10^7$ mol. This budget is dominated by transport fluxes (Inflow: $1.254 \times 10^9$ mol; Outflow: $-1.333 \times 10^9$ mol), whereas biological removal via denitrification accounts for only $-8.25 \times 10^6$ mol."

---

## Author Comment (AC1)

**Response to reviewers' comments on "Evaluating Long-Term Effectiveness of Managed Aquifer Recharge for Groundwater Recovery and Nitrate Mitigation in an Overexploited Aquifer System" by Y. Zhu, Z. Guo, S. Wan, K. Chen, Y. Wang, Z. Zeng, H. Shen, J. Ye, and C. Zheng**

Reviewer's comments in black; Response to reviewer's comments in blue; Revisions in the revised manuscript in red.

We would like to thank the editor and the reviewer for their constructive comments, which have helped us improve the presentation of this work. We have revised our manuscript according to the reviewer's comments and have provided below a point-by-point response to the reviewer's comments.

**Reviewer #1**

1. The paper is interesting and well written, and it is relevant to the water management of a vital area of China: Xiongan New District.

   There are two major issues that require a much more careful analysis, and I hope the authors can revise their paper accordingly. The first is the description of denitrification. In this manuscript, the authors used a two-step reduction to simplify the process. This is OK, but requires a much more careful and detailed analysis to justify its correctness. It is well known that denitrification is a complex process controlled by many factors. So, the authors should carefully justify why such a two-step reduction treatment is acceptable for this site!

   **Response:**

   Thank you for the detailed review. We agree that denitrification is a complex biogeochemical process involving multiple enzymatic pathways, intermediate products and microbial communities.

   The justification for adopting the two-step simplification ($NO_3^- \rightarrow NO_2^- \rightarrow N_2$) in our regional-scale model is based on the following three reasons:

   (1) The shallow aquifer system in the Xiong'an New Area is characterized by oxidizing conditions (high dissolved oxygen) and a scarcity of organic electron donors (Li et al., 2023). The Michaelis-Menten kinetics employed in our model (Eq. 5 and Table 1) explicitly include dual-Monod terms for electron donor limitation (Acetate as a proxy for DOC) and inhibition terms for dissolved oxygen. The overall denitrification rate is controlled primarily by the availability of the electron donor and the inhibition threshold of oxygen, rather than the transformation rates of intermediate species (like $N_2O$). Therefore, our model structure is sufficient to capture the "bottleneck" of the reaction at this site.

   (2) The primary objective of this study is to evaluate the long-term, regional-scale (536 km²) evolution of nitrate mass and the relative contribution of dilution versus reaction. The two-step reduction mechanism effectively conserves the mass balance of Nitrogen and simulates the permanent removal of

nitrate from the aqueous phase. While simplified, this approach is widely accepted in regional-scale Reactive Transport Modeling where computational efficiency is paramount, and parameter uncertainty for multi-step intermediate reactions would be prohibitively high (Guo et al., 2023; Karlović et al., 2022; Jin et al., 2024).

(3) Our results indicate that denitrification contributes only ~9% to the total nitrate reduction, while dilution dominates (~91%). Even if a more complex reaction network were used, the limiting environmental conditions, particularly the low availability of organic carbon and the high concentrations of dissolved oxygen, which collectively suppress denitrification rates. Thus, the current simplification provides a robust, albeit conservative, estimation of the biochemical contribution without over-parameterizing the model.

Revisions have been made in line 112:

"While denitrification involves complex enzymatic pathways, this two-step simplification is adopted based on the specific hydrogeochemical conditions of the study area, which is characterized by oxidizing environments and a scarcity of organic electron donors (Li et al., 2023). Under these conditions, the overall reaction rate is primarily controlled by the availability of electron donors and the inhibition threshold of dissolved oxygen, rather than the transformation rates of intermediate species. Therefore, the kinetic model employed Equation (5), which explicitly includes dual-Monod terms for donor limitation and oxygen inhibition, is sufficient to capture the rate-limiting steps and conserve nitrogen mass balance at the regional scale, while avoiding the excessive parameter uncertainty associated with complex multi-step reaction networks (Guo et al., 2023; Karlović et al., 2022; Jin et al., 2024)."

2. The second issue is related to Eq. 5. The author stated that " For example, according to the redox gradient theory, sulfate reduction may be inhibited by oxygen, nitrate, and trivalent iron dissolved in groundwater. These effects are modeled using an inhibition term in equation (5)". This is not sufficient. I would like the authors to explain what "inhibition term in Eq. (5)" is used and why. In one sentence: more elaboration is needed here!

**Response:**

Thank you for the detailed review. We apologize for the confusion caused by using "sulfate reduction" as a general example in the original text, which obscured the specific application to our nitrate model. In the revised manuscript, we will clarify that for the denitrification process modeled in this study, the "inhibition term" ($I$) in Equation (5) specifically refers to Oxygen Inhibition. As shown in Table 1 of our manuscript, the specific mathematical form used is:

$$I = \frac{K_{I,O_2}}{K_{I,O_2} + [O_2]}$$

The reason for using this specific inhibition term is based on the thermodynamic hierarchy of electron acceptors (Appelo and Postma, 2005). In groundwater systems, facultative anaerobes preferentially utilize Dissolved Oxygen (DO) over Nitrate as an electron acceptor because aerobic respiration provides a higher energy yield. Therefore, denitrification is strictly an anaerobic process that only occurs when

DO concentrations drop below a threshold. This inhibition term acts as a switch: high concentrations of DO will drive the term toward zero, effectively shutting down the denitrification reaction rate (R) in Equation (5) until hypoxic conditions are established.

We have rewritten the description following Equation (5) to remove the irrelevant example of sulfate reduction and explicitly state that oxygen is the sole inhibitor considered for the nitrate reduction pathway in this study.

Revisions have been made in line 117:

$$R = \mu_{max} \frac{C_{ED}}{C_{ED}+K_{ED}} \frac{C_{TEA}}{C_{TEA}+K_{TEA}} \frac{K_I}{K_I+C_I}$$

"Regarding the final term in Equation (5), it represents the inhibited effect controlled by thermodynamically more favorable electron acceptors. In the context of denitrification modeled here, the process is strictly anaerobic and is inhibited by dissolved oxygen (Appelo and Postma, 2005). Therefore, $C_I$ represents the concentration of dissolved oxygen. This term functions as a kinetic switch, reducing the reaction rate ($R$) effectively to zero when dissolved oxygen concentrations are high, reflecting the preferential utilization of oxygen over nitrate by facultative anaerobes."

3. L60, there should be a question mark "?" after the two questions mentioned there.

**Response:**

We have added a question mark in the revised manuscript.

Revisions have been made in line 60:

"1) How does MAR affect groundwater recovery in a severely depleted aquifer system? 2) How do heterogeneity and biogeochemical reactions interact with MAR to control the spatiotemporal evolution of nitrate concentrations in groundwater?"

4. L65, when NCP is mentioned for the first time, the full name of NCP must be provided.

**Response:**

Thank you for the detailed review. In our manuscript, the full name of NCP (North China Plain) is already provided at line 34 when it first appears.

Appelo, C. A. J. and Postma, D.: Geochemistry, groundwater and pollution. 2nd, Ed. Balkema, Rotterdam, 2005.

Guo, Z., Chen, K., Yi, S., and Zheng, C.: Response of groundwater quality to river-aquifer interactions during managed aquifer recharge: A reactive transport modeling analysis, Journal of Hydrology, 616, https://doi.org/10.1016/j.jhydrol.2022.128847, 2023.

Jin, Z., Tang, S., Yuan, L., Xu, Z., Chen, D., Liu, Z., Meng, X., Shen, Z., and Chen, L.: Areal artificial recharge has changed the interactions between surface water and groundwater, Journal of Hydrology, 637, 131318, https://doi.org/10.1016/j.jhydrol.2024.131318, 2024.

Karlović, I., Posavec, K., Larva, O., and Marković, T.: Numerical groundwater flow and nitrate transport assessment in alluvial aquifer of Varaždin region, NW Croatia, Journal of Hydrology: Regional Studies, 41, 101084, https://doi.org/10.1016/j.ejrh.2022.101084, 2022.

Li, N., Lyu, H., Xu, G., Chi, G., and Su, X.: Hydrogeochemical changes during artificial groundwater well recharge, Science of The Total Environment, 900, 165778, https://doi.org/10.1016/j.scitotenv.2023.165778, 2023.

---

## Author Comment (AC2)

**Response to reviewers' comments on "Evaluating Long-Term Effectiveness of Managed Aquifer Recharge for Groundwater Recovery and Nitrate Mitigation in an Overexploited Aquifer System" by Y. Zhu, Z. Guo, S. Wan, K. Chen, Y. Wang, Z. Zeng, H. Shen, J. Ye, and C. Zheng**

Reviewer's comments in black; Response to reviewer's comments in blue; Revisions in the revised manuscript in red.

We would like to thank the editor and the reviewer for their constructive comments, which have helped us improve the presentation of this work. We have revised our manuscript according to the reviewer's comments and have provided below a point-by-point response to the reviewer's comments.

**Reviewer #2**

This manuscript presents a regional-scale assessment of MAR impacts on groundwater level recovery and nitrate dynamics in the Xiong'an New Area using a three-dimensional coupled flow and reactive transport model. The topic is relevant for managing groundwater depletion and nitrate pollution in intensively cultivated regions. The integration of MAR hydraulics with nitrate biogeochemistry is an important contribution, and the authors provide clear insights into how dilution, denitrification, and geological heterogeneity jointly shape water-quality outcomes. The discussion is generally well-structured and highlights both the strengths and limitations of MAR. However, several aspects require clarification and improvement before the manuscript can be considered for publication. The model structure, boundary conditions, and parameterization of reactive processes need more transparency, particularly regarding organic carbon availability, redox controls, and the justification for simplifying nitrogen pathways. Some interpretations appear to overstate the certainty of denitrification estimates given the strong assumptions applied. While the authors have mentioned the influence of spatial resolution, recharge configuration, and heterogeneity, this part would benefit from more rigorous sensitivity analyses. The discussion could also more explicitly connect findings to broader MAR design principles and regional management implications. Furthermore, I suggest the authors discuss how their exclusion of other nitrogen transformation reactions might affect the conclusions regarding denitrification's minor role.

Overall, the study has clear potential, but refinements in methodological justification, uncertainty communication, and the contextualization of results will substantially strengthen the manuscript. Therefore, I recommend minor revisions.

**Response:**

Thank you for your encouraging feedback on the value of this work. We greatly appreciate your insights

and recommendations, which have notably improved the clarity and depth of our research. In response to your comments, we have made the necessary revisions. Below, we provide a point-by-point response to each comment and indicate the corresponding changes in the revised manuscript.

1. Line 16: The abstract states that "regional-scale hydrological and geochemical impacts remain insufficiently understood," but several recent MAR modeling papers at regional scales exist. It is unclear what is truly new: the model structure? the scale of the simulation? integrating heterogeneity? long-term simulation horizon? nitrate processes?

   **Response:**

   Despite recent advances in managed aquifer recharge (MAR) research using regional-scale models, studies that explicitly incorporate multi-component reactive transport processes under complex geological field conditions remain scarce. Accordingly, we have revised the abstract to more clearly articulate the specific contributions of this work.

   Revisions have been made in line l6:

   "Recent riverine MAR operations have shown significant local groundwater recovery; however, the long-term regional fate and spatial evolution of nitrate remain poorly quantified. In particular, it remains challenging to assess how geological heterogeneity interacts with biogeochemical processes to control remediation efficacy."

2. Line 13-24: Currently the abstract provides interpretation but no numbers, making the conclusions feel generic. Consider adding a sentence on the role of heterogeneity in shaping nitrate reduction patterns.

   **Response:**

   Thank you for this helpful comment. We have added a sentence on the role of heterogeneity in shaping nitrate reduction patterns, making the conclusions more specific and informative.

   Revisions have been made in line 15:

   "The results indicate that MAR leads to a basin-wide mean groundwater level rise of 1.11 m, with a maximum increase of 7.5 m near the river. Nitrate reduction is dominated by physical dilution (~91%) rather than denitrification (~9%). Furthermore, geological heterogeneity governs the spatial variability of water quality evolution by channeling flow through preferential pathways, which creates localized reduction hotspots, despite having a minimal impact on the total nitrate mass removal."

3. Line 33: "become as"

   **Response:**

   We have corrected the error by removing the redundant word "as" in the revised manuscript.

   Revisions have been made in line 33:

   "Amid these challenges, groundwater quality degradation, specially salinization and nitrate pollution, has become a pressing concern in many arid regions."

4. Line 113: The rate expression multiplies terms for electron donor (ED), terminal electron acceptor (TEA) and inhibition. However, there is no mention of an explicit microbial biomass pool (or active biomass concentration). Denitrification in aquifers is often biomass-mediated and can be substrate-limited and biomass-limited; excluding biomass (or at least an active biomass term) risks mischaracterizing dynamics, especially under long-term MAR where biomass may grow or be flushed

**Response:**

We sincerely appreciate the reviewer's insightful comment. We fully agree that denitrification is fundamentally a biomass-mediated process and that, in reality, reaction rates can be constrained by the active biomass concentration, particularly during the transient phases of microbial growth or decay under long-term MAR conditions. We employed an effective Monod-type formulation in this study, assuming a quasi-steady effective active biomass at the regional scale rather than explicitly simulating a dynamic microbial biomass pool. This decision is based on the following considerations:

(1) This is a regional-scale simulation (>500 km²). Introducing a dynamic biomass variable would require defining multiple unconstrained parameters (e.g., initial biomass distribution, yield coefficients, decay rates) that are unknown at this scale. Recent studies (Schäfer Rodrigues Silva et al., 2020) indicate that adding such unconstrained parameters often increases model uncertainty without significantly improving predictive capability for regional mass balance.

(2) Theoretical advances in reactive transport modeling suggest that at the macro-scale, complex micro-scale biological dynamics often collapse into effective kinetics driven by mixing and substrate availability. For instance, Le Traon (Le Traon et al., 2021) demonstrated that upscaled reaction rates are often controlled by physical transport limitations rather than intrinsic microbial physiology.

We have added a clarifying paragraph after Equation (5) to justify the simplified kinetic approach. While acknowledging this as a current limitation, we have further discussed the assumption in the revised manuscript. We agree that incorporating dynamic microbial growth is an important direction for future research, particularly for localized simulations near infiltration points, and will prioritize this in subsequent studies. We thank the reviewer again for highlighting this key mechanism, which has helped refine the boundary conditions and limitations of our present model.

Revisions have been made in line114:

"It is important to note that an explicit dynamic microbial biomass pool was not simulated due to the lack of spatial data for microbial parameters at the regional scale. Instead, a quasi-steady-state biomass concentration is assumed to be present in the aquifer sediments. Therefore, the effect of biomass abundance is implicitly incorporated into the effective rate constant ($\mu_{max}$). This simplification avoids the high parametric uncertainty associated with unconstrained biological parameters (Schäfer Rodrigues Silva et al., 2020) and aligns with theoretical frameworks for effective kinetics in heterogeneous media (Le Traon et al., 2021)."

Revisions have been made in line 374:

"Furthermore, the current model assumes a constant microbial capacity ($\mu_{max}$), neglecting potential biomass growth or washout during long-term recharge. It may overlook local biogeochemical dynamics,

such as biofilm development near infiltration zones, which warrants detailed investigation in future fine-scale studies."

5. Line 117: The Introduction criticizes prior studies for omitting microbial OM degradation and C–N coupling, but the method section does not convincingly demonstrate that these processes are represented.

**Response:**

Unlike many previous regional assessments that simplify denitrification as a first-order decay process, our model employs dual-Monod kinetics Equation (5). This formulation mathematically links the nitrate reduction rate to the concentration of the electron donor (Acetate). This kinetic dependence constitutes the actual "coupling" of Carbon and Nitrogen dynamics in our framework. Furthermore, we utilize Acetate ($CH_3COO^-$) as a representative proxy for bioavailable Dissolved Organic Carbon (DOC). The stoichiometric reaction in Table 1 explicitly simulates the consumption (degradation) of this organic matter concurrent with nitrate reduction.

Revisions have been made in line 52:

"Previous studies often simplify denitrification as a first-order decay process, thereby neglecting the kinetic coupling between nitrate reduction rates and the availability of electron donors."

6. Line 123: The manuscript specifies that the model domain is discretized using 100 m horizontal grid spacing but does not provide justification for this choice. Given the scale of the study area (tens of kilometers) and the strong emphasis on preferential flow paths, MAR-induced infiltration fronts, and heterogeneity, did the authors perform mesh and time-step sensitivity analyses (e.g., Courant number assessment or grid refinement tests) to ensure numerical stability and accuracy of both flow and reactive transport simulations?

**Response:**

We thank the reviewer for this valuable comment regarding grid resolution and numerical accuracy. To address this concern, we performed a grid resolution sensitivity analysis using three horizontal grid spacings (50 m, 75 m, and 100 m), while keeping all hydraulic parameters, boundary conditions, and recharge configurations identical.

[Figure]

**Figure R1. Grid-resolution sensitivity of simulated groundwater heads at representative points.: (a) Point 1 and (b) Point 2 using three horizontal grid resolutions (50 m, 75 m, and 100 m).**

The sensitivity analysis focused on the temporal evolution of the simulated groundwater head at representative observation points used in the manuscript. Figure R1 presents a comparison of groundwater head time series at two representative locations (Point 1 and Point 2) under the three grid resolutions. As shown in Fig. R1(a–b), the simulated groundwater head responses exhibit similar temporal patterns and comparable magnitudes across the tested grid spacings, indicating that the regional-scale hydraulic response is robust with respect to grid resolution within this range. Based on these results, the 100 m horizontal grid spacing was adopted for the full set of simulations, as it provides a reasonable balance between computational efficiency and spatial resolution for a regional-scale groundwater flow study covering a large domain.

Regarding temporal discretization, PFLOTRAN employs an adaptive time-stepping scheme for groundwater flow simulations, which automatically adjusts time steps to maintain numerical stability. No numerical instability or nonphysical oscillations were observed in the groundwater head simulations across the different grid resolutions.

Revisions have been made in line 123:

"The model domain (33 km × 26.9 km × 135 m) was discretized using a horizontal grid spacing of 100 m and three vertical layers with thicknesses of 15 m, 40 m, and 80 m, representing the vadose zone, the phreatic aquifer, and the underlying unit, respectively. A grid-resolution sensitivity test was conducted to ensure the numerical robustness of the groundwater flow simulation, based on which the 100 m horizontal grid spacing was adopted for the full set of simulations."

7. Line 129: Only four lithofacies are represented based on 14 boreholes. This is a very sparse conditioning dataset for a 3D domain of this size. The uncertainty introduced by such limited conditioning should be acknowledged.

**Response:**

We sincerely appreciate the reviewer for pointing out this critical limitation. We fully agree that representing a ~536 km² 3D domain based on only 14 boreholes constitutes a sparse conditioning dataset, which inevitably introduces structural uncertainty into the geological model. To explicitly address and bracket the uncertainty arising from this limited conditioning, we generated and simulated 20 realizations. The results presented in the manuscript are ensemble averages, which help mitigate the bias of any single realization and provide a probabilistic understanding of the flow field.

Revisions have been made in line 380:

"The geological heterogeneity was characterized based on a sparse borehole dataset, inevitably introducing structural uncertainty in the delineation of localized contaminant migration, although its impact is partially mitigated through stochastic ensemble simulations."

8.  Line 203: How sensitive are these values to recharge rates or hydraulic conductivity? The results repeatedly attribute attenuation to "distance and connectivity," but this remains qualitative. A clearer linkage between MAR response zones and the permeability map is needed.

**Response:**

We appreciate the reviewer's constructive comment. We agree that our original discussion on "distance and connectivity" was predominantly qualitative. To address this, we have performed a quantitative zonal analysis that explicitly links the MAR response ($\Delta h$) to the mapped hydrostratigraphic units.

[Figure]

**Figure R2. Distribution of Groundwater Head Recovery ($\Delta h$) Across Zones by 2035**

As illustrated in Figure R2, the statistical results reveal a stark contrast controlled by hydraulic conductivity:

(1) Zone 1 behaves as a hydraulic barrier, with 100% of its cells showing minimal recovery ($\Delta h < 1$ m), confirming that low permeability effectively blocks the propagation of the recharge signal. Similarly, the intermediate Zone 2 is dominated by weak responses (76% of cells < 1m), with only localized recovery

(mean rise ~2.23 m) in limited areas.

(2) In contrast, the permeable zones facilitate significant recovery. Zone 3 exhibits the most extensive response coverage, with over 64% of cells exceeding Δh >1m and one-third exceeding 3 m. Zone 4 (Highest Permeability), while spatially more focused (46% coverage), facilitates the most intense local recovery. It achieves the highest mean head rise (~3.18 m) and peak values (~7.5 m) among the responding cells.

This zone-by-zone quantification explicitly demonstrates that strong MAR benefits (Δh > 3 m) are confined to high-permeability corridors (Zones 3–4), while low-permeability zones remain largely unaffected.

Revisions have been made in line 208:

"A quantitative zonal analysis further confirms this control: Zone 1 acts as a hydraulic barrier with 100% of cells showing minimal recovery (Δh < 1 m). In contrast, the high permeability corridors facilitate significant recovery, with Zone 3 showing extensive coverage (64% cells > 1 m) and Zone 4 achieving the highest mean rise (~3.18 m) and peak values (~7.5 m). This demonstrates that strong MAR benefits are spatially confined to conductive pathways (Zones 3-4) rather than being uniformly distributed."

9. Line 215: The concept of comparing points with different distances and permeabilities is good, but the interpretation mixes permeability effects and distance effects. For example, line 220 states that "their amplitudes diverge with permeability: Point1 rises by ~4–7 m, while Point2 increases by ~3–5 m across the projection horizon". But figure 5b clearly shows a larger amplitude than 5a, which means MAR posed a higher effect on Point2. Also, no lag time between Point1&2 is surprising.

**Response:**

We thank the reviewer for this careful observation. We apologize for the confusion caused by our terminology.

(1) We would like to clarify that our analysis was based on monthly average water levels. The values mentioned in the text (rising by ~4–7 m vs. ~3–5 m) refer to the cumulative magnitude of water level rise (the net change over the simulation period), rather than the "seasonal fluctuation amplitude."

(2) The absence of a significant lag time between Point 1 and Point 2 is attributed to their immediate proximity to the recharge source. Both observation wells are located within < 500 m of the river channel. Given the rapid propagation of pressure waves in these near-field zones, the hydraulic response appears synchronous at the model's temporal resolution.

We have incorporated the updated figure into the main text and revised the corresponding descriptions accordingly.

Revisions have been made in line 220:

"Although the two sites respond nearly synchronously, their cumulative water-level rise: Point1 rises by ~4–7 m, while Point2 increases by ~3–5 m across the projection horizon (Fig. 5a–b)."

[Figure]

**Figure 5. Groundwater level changes and differences at observation points; (a)–(e) Time series of water level changes at observation points (Point 1 to Point 5) with and without MAR; (f) Differences in water levels (MAR vs Without MAR) for all observation points**

10. Line 255-256: Be cautious with statements such as "strongest decreases" when the absolute reductions remain quite small ($10^{-5}$ mol/L is extremely minor relative to typical groundwater nitrate levels). You may want to discuss ecological significance, not just the mathematical magnitude.

**Response:**

We sincerely thank the reviewer for this valid caution regarding the magnitude description. We agree that the term "strongest" could be misleading, given the small absolute reduction. Instead of adding a lengthy discussion, we have refined the description directly in Section 3.3 to provide immediate context and avoid overstatement.

Revisions have been made in line 255:

"The response classes correspond to orders of magnitude: -8 to -7 (very small decrease), -7 to -6 (small),

-6 to -5 (moderate to large), and > -5 (the most distinct decreases, at least on the order of $10^{-5}$), while the absolute magnitude ($10^{-5}$) appears minor, it represents the primary zones of cumulative mass removal in the system."

11. Line 278: Obviously, denitrification is effectively negligible in this system. How reliable this result is? Plus, it is surprising that the spatial pattern is nearly uniform, given strong variability in residence time, electron donor availability, and organic carbon distribution in real systems.

**Response:**

We appreciate this insightful comment. Although residence time usually drives reaction patterns, the hydrogeochemical constraints (high oxygen, low carbon) in our study area suppress denitrification basin-wide. This strong chemical limitation effectively masks the influence of physical heterogeneity, resulting in a uniform and negligible reaction rate.

(1) The result is reliable because it is constrained by site-specific monitoring data used as model inputs (Table 2). The shallow groundwater system in Xiong'an is characterized by oxidizing conditions and carbon limitation. According to the reaction kinetics Equation (5), this high oxygen level exerts a strong inhibition effect. Consequently, the system is thermodynamically unfavorable for denitrification. The model correctly predicts that nitrate removal is dominated by physical dilution (~91%) rather than biological reduction (~9%) under these aerobic conditions.

(2) The reviewer correctly notes that variability in residence time usually creates heterogeneous reaction patterns. However, this applies primarily when the reaction is active. In our system, the reaction is globally suppressed by the widespread presence of oxygen and the scarcity of electron donors. Since the oxygen inhibition is applied uniformly strong across the domain, the reaction rate is clamped to near-zero levels everywhere. This chemical limitation acts as the dominant control, masking the secondary effects of physical heterogeneity (residence time).

Revisions have been made in line 285:

"The modeled negligible and uniform denitrification is fundamentally driven by the hydrogeochemical regime of the study site. Field data (Table 2) indicates that the aquifer is generally oxidizing and carbon-limited. Under such conditions, the reaction kinetics are globally suppressed by oxygen inhibition and electron donor starvation. Consequently, the influence of physical heterogeneity (i.e., variability in residence time) becomes secondary to this overwhelming chemical limitation. Even in zones with long residence times, the reaction rates remain low due to the lack of favorable redox conditions, resulting in a spatially uniform distribution of minimal denitrification."

12. Line 353: You highlight the differing effects of heterogeneity on dilution vs. denitrification but do not clearly link these insights to the broader literature on preferential flow and residence-time control on reactive transport. Adding comparison to previous heterogeneity–redox studies would position your results more convincingly.

**Response:**

We sincerely thank the reviewer for this constructive suggestion. In the revised Discussion, we have explicitly interpreted the differing effects of heterogeneity through the lens of groundwater residence times and contact time. We clarified that high-permeability zones act as preferential flow paths, which drastically increase flow velocity and reduce the time groundwater spends in the aquifer. As established in classical reactive transport literature (Zheng and Gorelick, 2003), when solutes are transported rapidly through the system, the effective contact time between nitrate and reactive microbial communities is insufficient for substantial biodegradation to occur.

Revisions have been made in line 369:

"This aligns with previous studies (Zheng and Gorelick, 2003; Green et al., 2010; Chen et al., 2024), which indicate that while high-preferential flow enhances physical dilution via dispersion, they simultaneously restricts denitrification by shortening groundwater residence times. Consequently, rapid transport results in insufficient effective contact time for substantial biodegradation to occur."

13. Line 362: "he"

    **Response:**

    We have corrected "he" to "The" in the revised manuscript.

14. Line 373: While you note that the recharge scheme assumed constant conditions, you do not discuss how sensitive the nitrate response is to recharge timing, distribution, or water chemistry. Without acknowledging potential model sensitivity, the management implications may appear overstated.

    **Response:**

    We thank the reviewer for this crucial observation. We fully agree that assuming constant recharge conditions simplifies the complexity of real-world operations, where seasonal timing, spatial distribution, and hydrochemistry vary significantly. Neglecting these factors could indeed lead to overstated management implications.

    Revisions have been made in line 376:

    "The simulated nitrate reduction represents a baseline scenario under constant recharge. In practice, operational intermittency (wetting-drying cycles) could enhance oxygen intrusion into the vadose zone, strengthening denitrification inhibition. Furthermore, fluctuations in source water chemistry, particularly reductions in Dissolved Organic Carbon (DOC) load, would limit electron donor availability. Consequently, actual nitrate mitigation efficiency may be lower than simulated if hydraulic saturation and sufficient carbon supply are not maintained."

    Revisions have been made in line 394:

    "Overall, MAR proves to be a robust tool for hydraulic recovery; however, its effectiveness for nitrate mitigation is primarily driven by physical dilution. Therefore, its implementation as a remediation strategy requires careful consideration of source water chemistry and continuous injection regimes to maximize benefits."

Chen, K., Guo, Z., Zhan, Y., Roden, E. E., and Zheng, C.: Heterogeneity in permeability and particulate organic carbon content controls the redox condition of riverbed sediments at different timescales, Geophysical Research Letters, 51, e2023GL107761, https://doi.org/10.1029/2023GL107761, 2024.

Green, C. T., Böhlke, J. K., Bekins, B. A., and Phillips, S. P.: Mixing effects on apparent reaction rates and isotope fractionation during denitrification in a heterogeneous aquifer, Water resources research, 46, https://doi.org/10.1029/2009WR008903, 2010.

Le Traon, C., Aquino, T., Bouchez, C., Maher, K., and Le Borgne, T.: Effective kinetics driven by dynamic concentration gradients under coupled transport and reaction, Geochimica et Cosmochimica Acta, 306, 189–209, https://doi.org/10.1016/j.gca.2021.04.033, 2021.

Schäfer Rodrigues Silva, A., Guthke, A., Höge, M., Cirpka, O. A., and Nowak, W.: Strategies for simplifying reactive transport models: A Bayesian model comparison, Water Resources Research, 56, e2020WR028100, https://doi.org/10.1029/2020WR028100, 2020.

Zheng, C. and Gorelick, S. M.: Analysis of solute transport in flow fields influenced by preferential flowpaths at the decimeter scale, Groundwater, 41, 142–155, https://doi.org/10.1111/j.1745-6584.2003.tb02578.x, 2003.

---

## Author Response (AR2)

**Response to reviewers' comments on "Evaluating Long-Term Effectiveness of Managed Aquifer Recharge for Groundwater Recovery and Nitrate Mitigation in an Overexploited Aquifer System" by Y. Zhu, Z. Guo, S. Wan, K. Chen, Y. Wang, Z. Zeng, H. Shen, J. Ye, and C. Zheng**

Reviewer's comments in black; Response to reviewer' s comments in blue; Revisions in the revised manuscript in red.

We would like to thank the editor and the reviewer for their constructive comments, which have helped us improve the presentation of this work. We have revised our manuscript according to the reviewer's comments and have provided below a point-by-point response to the reviewer's comments.

**Reviewer #1**

The authors addressed my comments satisfactorily. They provide a detailed response to my first question in review, but they did not include their detailed response in the revised text. Instead, they presented a shorter version there.I will suggest they include a detailed explanation in the revised text, rather than the shorter version, as this is an important issue. A minor revision is required.

**Response:**

Thank you for the suggestion. We have incorporated the detailed explanation from our previous response into the revised manuscript as requested.

Revisions have been made in lines l21-134:

"Although denitrification inherently involves complex enzymatic pathways and intermediate products, a simplified two-step reduction scheme was adopted in this study based on the specific hydrogeological context of the Xiong'an New Area. First, the shallow aquifer system is characterized by predominantly oxidizing conditions, as evidenced by high dissolved oxygen concentrations and a scarcity of organic electron donors (Li et al., 2023). Under such biogeochemical constraints, the overall reaction rate is governed primarily by the availability of electron donors and the inhibition threshold of dissolved oxygen, rather than by the transformation kinetics of intermediate species. Reflecting this, the kinetic formulation in Equation (5) incorporates dual-Monod terms to explicitly represent donor limitation and oxygen inhibition, effectively capturing the rate-limiting steps. Second, the regional-scale and long-term nature of the simulation (536 km$^2$) necessitates a balance between mechanistic detail and computational feasibility. The two-step reduction scheme ensures the conservation of nitrogen mass balance while maintaining numerical efficiency, avoiding the excessive parameter uncertainty associated with complex multi-step reaction networks. This approach aligns with established practices in regional reactive transport modeling (Guo et al., 2023; Karlović et al., 2022; Jin et al., 2024)."

Revisions have been made in lines 331-333:

"This outcome validates the suitability of the adopted two-step reduction scheme (Equation 5), which effectively captures the suppression of denitrification rates under such donor-poor and oxidizing conditions."

Revisions have been made in lines 416-418:

"These field observations justify the exclusion of complex multi-step reaction networks in our model, as the scarcity of primary electron donors constitutes the rate-limiting step rather than intermediate transformation pathways."

**References**

Guo, Z., Chen, K., Yi, S., and Zheng, C.: Response of groundwater quality to river-aquifer interactions during managed aquifer recharge: A reactive transport modeling analysis, Journal of Hydrology, 616, 128847, https://doi.org/10.1016/j.jhydrol.2022.128847, 2023.

Jin, Z., Tang, S., Yuan, L., Xu, Z., Chen, D., Liu, Z., Meng, X., Shen, Z., and Chen, L.: Areal artificial recharge has changed the interactions between surface water and groundwater, Journal of Hydrology, 637, 131318, https://doi.org/10.1016/j.jhydrol.2024.131318, 2024.

Karlović, I., Posavec, K., Larva, O., and Marković, T.: Numerical groundwater flow and nitrate transport assessment in alluvial aquifer of Varaždin region, NW Croatia, Journal of Hydrology: Regional Studies, 41, 101084, https://doi.org/10.1016/j.ejrh.2022.101084, 2022.

Li, N., Lyu, H., Xu, G., Chi, G., and Su, X.: Hydrogeochemical changes during artificial groundwater well recharge, Science of The Total Environment, 900, 165778, https://doi.org/10.1016/j.scitotenv.2023.165778, 2023.

**Reviewer #2**

My previous concerns have been well addressed but some minor language errors have been noticed which can be handled in publication procedure. Overall, I recommend acceptance for publication and suggest careful language check.

**Response:**

Thank you for your positive feedback and the recommendation for acceptance. We have carefully proofread the manuscript again to correct potential language errors as suggested.